# Genetic, Antigenic, and Pathobiological Characterization of H9 and H6 Low Pathogenicity Avian Influenza Viruses Isolated in Vietnam from 2014 to 2018

**DOI:** 10.3390/microorganisms11020244

**Published:** 2023-01-18

**Authors:** Kien Trung Le, Lam Thanh Nguyen, Loc Tan Huynh, Duc-Huy Chu, Long Van Nguyen, Tien Ngoc Nguyen, Tien Ngoc Tien, Keita Matsuno, Masatoshi Okamatsu, Takahiro Hiono, Norikazu Isoda, Yoshihiro Sakoda

**Affiliations:** 1Laboratory of Microbiology, Department of Disease Control, Faculty of Veterinary Medicine, Hokkaido University, Sapporo 060-0818, Japan; 2Department of Veterinary Medicine, College of Agriculture, Can Tho University, Can Tho 941-15, Vietnam; 3Department of Animal Health, Ministry of Agriculture and Rural Development, Ha Noi 115-19, Vietnam; 4Regional Animal Health Office VII, Department of Animal Health, Ministry of Agriculture and Rural Development, Can Tho 942-07, Vietnam; 5Division of Risk Analysis and Management, International Institute for Zoonosis Control, Hokkaido University, Sapporo 001-0020, Japan; 6One Health Research Center, Hokkaido University, Sapporo 060-0818, Japan; 7International Collaboration Unit, International Institute for Zoonosis Control, Hokkaido University, Sapporo 001-0020, Japan; 8Hokkaido University Institute for Vaccine Research and Development (HU-IVReD), Hokkaido University, Sapporo 001-0021, Japan

**Keywords:** coinfection, *Escherichia coli* O2, low pathogenicity avian influenza, pathogenicity assessment, surveillance, Vietnam

## Abstract

The H9 and H6 subtypes of low pathogenicity avian influenza viruses (LPAIVs) cause substantial economic losses in poultry worldwide, including Vietnam. Herein, we characterized Vietnamese H9 and H6 LPAIVs to facilitate the control of avian influenza. The space–time representative viruses of each subtype were selected based on active surveillance from 2014 to 2018 in Vietnam. Phylogenetic analysis using hemagglutinin genes revealed that 54 H9 and 48 H6 Vietnamese LPAIVs were classified into the sublineages Y280/BJ94 and Group II, respectively. Gene constellation analysis indicated that 6 and 19 genotypes of the H9 and H6 subtypes, respectively, belonged to the representative viruses. The Vietnamese viruses are genetically related to the previous isolates and those in neighboring countries, indicating their circulation in poultry after being introduced into Vietnam. The antigenicity of these subtypes was different from that of viruses isolated from wild birds. Antigenicity was more conserved in the H9 viruses than in the H6 viruses. Furthermore, a representative H9 LPAIV exhibited systemic replication in chickens, which was enhanced by coinfection with avian pathogenic *Escherichia coli* O2. Although H9 and H6 were classified as LPAIVs, their characterization indicated that their silent spread might significantly affect the poultry industry.

## 1. Introduction

Avian influenza viruses (AIVs), belonging to the genus *Alphainfluenzavirus* and family *Orthomyxoviridae*, cause avian influenza (AI). The genome of AIVs comprises eight negative-sense single-stranded RNA segments [1]. Based on the antigenic characteristics of the surface glycoproteins, AIVs are classified into 16 hemagglutinin ([HA], H1–H16) and 9 neuraminidase ([NA], N1–N9) subtypes [2,3]. All subtypes have been detected in wild aquatic birds, the natural reservoirs of AIVs. Recently, two novel influenza A viruses, H17N10 and H18N11, were genetically detected in bats [4]. Based on the pathogenicity of these viruses in chickens, AIVs are categorized into high pathogenicity AIVs (HPAIVs) and low pathogenicity AIVs (LPAIVs). Although monitoring clinical signs and laboratory diagnosis can help identify an HPAIV outbreak, LPAIV infection usually causes asymptomatic disease. Therefore, an undiagnosed LPAIV infection might result in an LPAI outbreak being overlooked. LPAIV infection has caused global economic losses in poultry farms, affecting egg and meat production [5,6]. Unfortunately, in many countries, including Vietnam, an LPAI outbreak is easily overlooked, hindering further analysis of necessary data. As culling is not mandatory for LPAI outbreaks under the regulation of the World Organization for Animal Health (WOAH), there is no effective strategy for controlling LPAIV infections.

Among the AIV subtypes detected in Asia, the H9 and H6 subtypes are predominant in wild birds and poultry [7,8]. Although AIVs of these subtypes are classified as LPAIVs, which cause asymptomatic infection in poultry, they can still potentially cause severe respiratory distress due to the occasional coinfection with other pathogens [9,10,11]. Additionally, LPAIVs, including H9 and H6 subtypes, can lead to the development of potential pandemic strains via reassortment [12]. Except for the H5 and H7 subtypes, reports have shown the zoonotic potential of other LPAIVs, including H6N1 [13] and H9N2 [14,15]. Therefore, the Vietnamese government implemented many interventions to minimize the transmission of AIVs in poultry.

Since 2009, the WOAH Regional Representation for Asia and the Pacific has supported the active surveillance program to control and prevent AIV infections in poultry in Vietnam [16,17,18,19,20]. Along with several H9 and H6 LPAIV strains isolated from poultry during this surveillance program, many bacterial strains, including avian pathogenic *Escherichia coli* (APEC), were isolated from the same flock of domestic poultry. APEC has been reported to cause severe diseases in humans and animals due to the presence of virulence factors in hosts [21]. APEC belongs to the zoonotic potential group of *E. coli*, which includes the serotypes O1, O2, O18, and O78. These are the most common APEC strains, accounting for over 80% of the infections [22]. Therefore, the co-circulation of these two pathogens seriously threatens both animals and humans. The present study reports the genetic and antigenic evolution of H9 and H6 LPAIVs isolated in Vietnam. The synergistic effect of APEC coinfection on the pathogenicity of H9 LPAIV was evaluated via animal experiments.

## 2. Materials and Methods

### 2.1. Sample Collection

From 2014 to 2018, the surveillance for AI was conducted in three Vietnamese provinces, Lang Son, Hue, and Vinh Long (Appendix A). Oropharyngeal, cloacal, and environmental swabs were obtained from domestic birds housed at biosecurity or backyard farms, live bird markets, and poultry delivery stations. The field samples were preserved in a viral transport medium (VTM), which is a minimum essential medium containing penicillin, streptomycin, gentamicin, nystatin, and bovine serum. All samples were stored at −80 °C until virological tests. Influenza type A virus was screened by pooling up to 10 swab samples and then subjected to RT-PCR targeting on M gene following the WOAH’s instructions [23]. After screening in Vietnam, all samples were transferred to the Laboratory of Microbiology, Faculty of Veterinary Medicine, Hokkaido University, Japan, to isolate the infectious viruses.

### 2.2. Isolation and Identification of AIVs

Each sample in an M-gene-positive pooled sample was resuspended in VTM and inoculated into the allantoic cavity of 10-day-old embryonated chicken eggs. After incubation at 35 °C for 30 to 48 h, the allantoic fluid was collected for HA assay to detect the virus. The isolated influenza virus was subtyped using HA inhibition (HI) and NA inhibition tests with antisera against the reference influenza virus strains [24].

### 2.3. Sequencing and Phylogenetic Analysis

The representative isolates were selected based on space–time and species distribution. Two hundred and fifty microliters of allantoic fluid was used to extract viral RNA using TRIzol LS Reagent (Thermo Fisher Scientific, Waltham, MA, USA) according to the manufacturer’s protocol, followed by reverse transcription using the Uni12 primer [25] and Moloney Murine Leukemia Virus Reverse Transcriptase (Thermo Fisher Scientific). Full-length complementary DNAs (cDNAs) of the HA gene segments were amplified using PCR with Ex-Taq (Takara Bio, Kusatsu, Japan) and gene-specific primer sets [25]. Direct sequencing of the HA gene segment was performed using the BigDye Terminator v3.1 Cycle Sequencing Kit and on the 3500 Genetic Analyzer (Thermo Fisher Scientific). The full genome sequence was confirmed using next-generation sequencing. MiSeq libraries were prepared using the NEBNext Ultra RNA Library Prep Kit for Illumina (New England Biolabs, Ipswich, MA, USA) and sequenced using the MiSeq system and MiSeq Reagent Kit v3 (600 cycles) (Illumina, San Diego, CA, USA). The sequence reads were assembled using CLC Genomics Workbench, version 12 (CLC Bio, Aarhus, Denmark; now Qiagen). The deduced amino acid sequence of HA was interpreted from the cDNA sequence information using GENETYX version 12 (Genetyx Corporation, Tokyo, Japan).

To investigate the genetic relationship, the nucleotide sequences of the representative strains were aligned with those from a public database using Clustal W version 2.0 [26]. The maximum likelihood method, with 1000 bootstrap replicates, was applied to construct the phylogenetic tree using MEGA 7.0 software (version 7.0.26, University of Kent, Canterbury, UK) [27].

### 2.4. Cross-HI Test and Antigenic Cartography

Polyclonal antisera were obtained from hyperimmune chickens against reference H9 (A/duck/Hong Kong/Y280/1997 (H9N2), A/quail/Hong Kong/G1/1997 (H9N2), A/duck/Hokkaido/49/1998 (H9N2) (Hok/49/98), and A/turkey/Wisconsin/1/1966 (H9N2) (Wis/1/56)) or H6 (A/duck/Hong Kong/960/1980 (H6N2), A/teal/Hong Kong/W312/1997 (H6N1), A/duck/Vietnam/OIE-4429/2010 (H6N2), A/duck/Hokkaido/262/2004 (H6N1), A/shearwater/Australia/1/1972 (H6N5), and A/turkey/Massachusetts/3740/1965 (H6N2)) subtypes as previously described [28]. The antigenic properties of the representative viruses were assessed using polyclonal antisera using a cross-HI test as previously described [29].

Using web-based software, the resulting data containing cross-HI titers were used to estimate the *x*/*y* coordinates of each antiserum and antigen in the antigenic cartography [30]. In an antigenic map, both vertical and horizontal axes represent the antigenic distance. One antigenic-unit distance corresponds to a 2-fold dilution in the HI assay. More than two antigenic units were regarded as a significant difference.

### 2.5. Isolation and Identification of E. coli

During the surveillance, trachea swabs collected from the poultry were stored in soft nutrient agar until bacterial isolation. The trachea swabs were cultured in nutrient broth (FUJIFILM Wako, Osaka, Japan) and incubated at 37 °C for 24 h for enrichment. The potential isolates were identified by plating the enriched samples in MacConkey agar (Nissui Pharmaceutical, Tokyo, Japan) and incubated at 37 °C for 24 h. DNA was extracted from the selected isolates using the heat shock method and then subjected to conventional PCR to determine the *E. coli* serotype. Primer pairs were used to detect and differentiate serotypes O1, O2, O18, and O78 as previously described [31]. The genes related to the pathogenicity of *E. coli* in poultry were identified by genotyping the isolates for *papC*, *fimC*, *tsh*, *iucD* [32], *astA*, *iutA*, *irp2* [33], *cva/cviC* [34], and *iss* [35] (Appendix A). Of all the isolated *E. coli* strains, the serotype with the highest accumulation of virulence-associated genes was selected for animal experiments.

### 2.6. Animal Experiments

All chickens were hatched in our laboratory from the embryonated eggs of a conventional chicken flock that was free of AIV antibodies. *E. coli* is persistent in the environment and gastrointestinal tract of poultry. This evidence was widely agreed upon in previous studies [36,37,38]. Thus, the most plausible scenario for *E. coli* and AIV coinfection in the field is that the poultry were first infected by *E. coli* and then by AIVs. In this regard, the animal experiment was designed as follows: 4-week-old experimental chickens were categorized into the negative control (*n* = 3), bacterial infection (*E. coli* alone, *n* = 6), viral infection (H9N2 alone, *n* = 6), and coinfection (*E. coli* and H9N2, *n* = 6) groups. *E. coli* and H9 LPAIVs were inoculated into experimental chickens intranasally. The chickens in the negative control group were inoculated with PBS. The chickens in the bacterial and viral infection groups were inoculated with 10^9.0^ colony-forming units (CFUs) of *E. coli* per head and 10^6.0^ egg infectious doses (EID)_50_/head of H9N2 LPAIV, respectively. The chickens in the coinfection group were inoculated with 10^9.0^ CFU/head of *E. coli* 2 days prior to H9N2 challenge (10^6.0^ EID_50_/head).

The chickens were monitored for clinical signs every day up to 14 days post inoculation (dpi). The general condition of chickens was recorded based on the clinical signs of disease and mortality. Any type of pathology was recorded, especially respiratory abnormalities, such as nasal discharge, sneezing, tracheal rales, coughing, difficulty breathing, and head swelling or depression. A scoring system was used to evaluate the severity of clinical signs, which were scored on the following scale: 0—no sign, 1—sick (one of the abovementioned signs was observed), 2—severely sick (two or more of the abovementioned signs were observed), and 3—dead. The average clinical score was calculated using the sum of clinical scores for each chicken divided by the number of chickens in each group at each observation time. To determine the virus recovery in the organs, the brain, trachea, lung, spleen, liver, pancreas, kidney, colon, and blood in citric acid were collected at 3 dpi. The tracheal and cloacal swabs were collected every day from 1 to 14 dpi to check for virus shedding.

All animal experiments were performed in the animal biosafety level 3 facility at the Faculty of Veterinary Medicine of Hokkaido University, Sapporo, Japan (approval number: 18-0037, obtained on 3 March 2019) following the guidelines of the Institutional Animal and Committee of Hokkaido University, certified by the Association for Assessment and Accreditation of Laboratory Animal Care International (AAALAC International) since 2007.

### 2.7. Virus Titration

For the collected internal organs, Multi-Beads Shocker (Yasui Kikai, Osaka, Japan) was used to prepare 10% (*w*/*v*) homogenate of the organs with the VTM. The suspension was centrifuged at 2000 rpm for 10 min to obtain its supernatant. Then, the supernatant was serially diluted 10 times and each diluted aliquot was inoculated in four 10-day-old embryonated chicken eggs. The viral growth was confirmed in terms of HA activity after incubating the samples at 35 °C for 48 h. The viral recovery titers were calculated as EID_50_.

### 2.8. Statistical Analysis

To compare the viral growth between the groups, one-way ANOVA with Tukey’s test was performed using the R software (version 4.1.0, R Foundation for statistical computing, Vienna, Austria) [39]. Differences with a *p*-value of <0.05 were considered statistically significant.

## 3. Results

### 3.1. Identification of the AIVs Circulating in Poultry

From 2014 to 2018, 1361 viruses were isolated from 15431 cloacal and oropharyngeal samples of domestic birds and environmental samples (Appendix A), indicating an AIV prevalence of 8.8% (95% confidence interval: 8.4–9.3). The virological characterization were conducted for some H6 and H9 viruses isolated in Hue province in 2014, H7 viruses isolated in Vinh Long province in 2018, and H5 HPAIVs isolated from 2014 to 2017 in previous studies [18,29,40]. The other isolates were reported only in an epidemiological aspect [19,41] or newly characterized in the present study. Based on HA classification, 69 of H3, 8 of H4, 213 of H5, 344 of H6, 3 of H7, 698 of H9, 9 of H10, 15 of H11, 1 of H12, and 1 of H13 AIVs were identified, while based on NA classification, 217 of N1, 771 of N2, 2 of N3, 1 of N5, 348 of N6, 7 of N7, 5 of N8, and 10 of N9 AIVs were identified. A total of 1148 LPAIVs (accounting for 84.3% of the total number of isolates) were identified, wherein the H9 subtype was predominant (60.8%), followed by the H6 subtype (30.0%). The HPAIVs accounted for only 15.7% of the total number of isolates.

### 3.2. Phylogenetic Analysis of the HA and NA Genes of H9 and H6 LPAIVs

Phylogenetic analysis was performed to assess the genetic relationship between the Vietnamese isolates and those in neighboring countries. The H9 HA genes were found to be phylogenetically composed of two lineages: Eurasian and North American. The H9 viruses isolated based on this surveillance were clustered into Clade 15 of the Y280/BJ94 sublineage in the Eurasian lineage. These viruses were genetically related to viruses isolated previously from North Vietnam in 2012 and those isolated from poultry in China between 1997 and 2012 (Figure 1).

Based on the HA genes, the H6 viruses were phylogenetically classified into two lineages: Eurasian and North American. Furthermore, the Eurasian lineage was divided into five distinct sublineages: Group I, Group II, Group III, W312, and Early, as previously described [16,17]. Most H6 viruses isolated in this surveillance were clustered into Group II. Three viruses were clustered into Group III with the viruses isolated in Vietnam in 2012 (Figure 2).

The NA gene segments were classified based on previous studies [16,17]. While the classification of N2 NA genes of the H9 LPAIVs and N6 NA genes of the H6 LPAIVs was same as that of the HA gene, the N2 NA genes of the H6 LPAIVs were classified into Group II (Appendix A).

### 3.3. Genotyping the Isolated H9 and H6 LPAIVs

Phylogenetic analysis was performed using six internal gene segments of the AIVs isolated in Vietnam to investigate the genetic diversity of the currently circulating AIVs (Appendix A–H). The names of the genetic groups for each internal gene were defined in previous studies [16,17]. The six internal gene segments were classified into H6 Group I, Group II, Group III, W312, JX8264-like, Vietnam, Hunan491-like, Y280/BJ94, H9 China, wild birds, and Gs/Gd-like. Among the representative Vietnamese H9 viruses, six genotypes were identified with a dominant genotype (accounting for 61% of the representative H9 viruses). A total of 19 genotypes were identified among the representative H6 viruses but none of them were predominant (Figure 3). These results indicated that the genetic diversification of the Vietnamese H6 viruses was higher than that of the H9 viruses.

### 3.4. Antigenic Analysis of the H9 LPAIVs

To examine changes in the antigenicity of H9 viruses, isolates from 2009 to 2012 were antigenically analyzed using the cross-HI test. Seven Vietnamese representative H9 viruses were analyzed (Appendix A). All the Vietnamese isolates in this study strongly reacted with the antisera of A/duck/Hong Kong/Y280/1997 (H9N2) and A/chicken/Vietnam/HU8-1860/2017 (H9N2), belonging to the Y280/BJ94 sublineage. Similarity, all Vietnamese isolates showed a strong reaction with the antisera of Hok/49/98 (H9N2) (Y439 sublineage). However, these viruses showed a moderate reaction with the antisera of HK/G1/97 (H9N2) (G1 sublineage) and weak reaction with the antisera of Wis/1/56 (H9N2) (North American lineages). The antigenic map developed using the cross-HI test results indicated that the representative H9 viruses were antigenically distant from the wild bird antigenic group (Hok/49/98 and Wis/1/56). Moreover, in terms of antigenicity, the H9 LPAIVs did not diverge and formed the same cluster with other viruses from domestic birds (Figure 4). These results suggested that the antigenicity of the Vietnamese H9 viruses was stable during their circulation in poultry.

### 3.5. Antigenic Analysis of the H6 LPAIVs

Isolates obtained in 2010 were used for the cross-HI test to evaluate the antigenic drift. Ten Vietnamese representative H6 strains were selected for antigenic analysis by cross-HI test using a panel of chicken antisera against eight viruses of different sublineages (Appendix A). Most representative H6 viruses showed a weak reaction with the antisera of A/duck/Vietnam/OIE-4429/2010 (H6N2) but a strong reaction with that of A/duck/Vietnam/HU1-637/2014 (H6N6) and A/duck/Vietnam/HU8-1088/2017 (H6N6), although they were all classified into the Group II sublineage. Despite the classification of A/duck/Vietnam/HU4-906/2015 (H6N6) and A/duck/Hokkaido/262/2004 (H6N1) viruses in Group III, we observed a weak reaction between A/duck/Vietnam/HU4-906/2015 (H6N6) virus and the antisera of A/duck/Hokkaido/262/2004 (H6N1). The cross-HI test results were used for antigenic cartography. The antigenic map indicated that the representative H6 viruses were mainly classified into two antigenic groups (one formed by A/duck/Vietnam/OIE-4429/2010 and another by A/duck/Vietnam/HU8-1088/2017), which were independent of the wild bird antigenic group (Figure 5). These results implied that the H6 viruses isolated in this study have undergone antigenic diversification in the poultry population.

### 3.6. Virulence-Associated Genes of APEC

The serogroups O1 (1 strain), O2 (1 strain), O18 (1 strain), and O78 (2 strains) were identified among 23 strains of *E. coli* isolated during the 2019 surveillance. Analysis of virulence-associated genes indicated that *E. coli* O2 had the highest number of virulence-associated genes (seven genes), followed by *E. coli* O1 and O18 with six genes and *E. coli* O78 with five genes (Appendix A). The existence of the type 1 fimbriae gene (*fimC*), temperature-sensitive hemagglutinin (*tsh*), aerobactin gene (*iucD*), enteroaggregative toxin gene (*astA*), increased serum survival gene (*iss*), iron repressible gene (*irp2*), and iron transport gene (*iutA*) in the *E. coli* O2 strain may be a higher potential risk factor for infection in poultry than the existence of those in other *E. coli* isolates.

### 3.7. Pathogenicity of H9N2 LPAIVs in the Coinfection with E. coli in Chickens

The chickens in the negative control group did not show any clinical signs throughout the experimental period. A few minor clinical signs, including ruffled feathers, hunched posture, or depression, were observed in two chickens in the bacterial infection group on 1 dpi. While chickens infected with H9N2 LPAIV alone did not exhibit any clinical signs, those coinfected with *E. coli* O2 and H9N2 showed more clinical signs at 1–2 dpi than chickens infected with *E. coli* O2 alone. However, these signs disappeared from 3 dpi onward (Appendix A).

### 3.8. Virus Recovery from the Chicken Organs

The virus was recovered from several organs of the chickens in the viral infection group (A/chicken/Vietnam/HU8-1860/2017–H9N2 alone), including the brain, trachea, lung, spleen, liver, kidney, colon, and blood, which revealed systemic replication of H9N2 LPAIV isolated in Vietnam. However, higher viral titers were found in the respiratory organs (Figure 6). In the coinfection group (*E. coli* O2 and A/chicken/Vietnam/HU8-1860/2017–*E. coli* + H9N2), the viral titers were significantly higher in the internal organs (spleen, pancreas, kidney, and colon) and blood compared with those in the viral infection group, implying the synergistic effect of *E. coli* O2 and H9N2 LPAIV coinfection.

### 3.9. Virus Shedding during the Experiment

To determine the difference in virus shedding between the virus infection (A/chicken/Vietnam/HU8-1860/2017–H9N2 alone) and coinfection groups (*E. coli* O2 and A/chicken/Vietnam/HU8-1860/2017–*E. coli* + H9N2), swabs were collected from 1 to 14 dpi and inoculated to chicken embryonated eggs. The virus titers were significantly higher in both trachea and cloacal swabs of the chicken in the coinfection group than those in the virus infection group from 1 to 6 dpi (Figure 7). The peak of virus shedding was earlier, and the shedding period was relatively longer in the coinfection group compared to the virus infection group, suggesting that coinfection with *E. coli* O2 might promote the virus-shedding process by enhancing and prolonging this process.

## 4. Discussion

To the best of our knowledge, this study is the first to comprehensively assess the characteristics of LPAIVs in Vietnam. The link between the genomic data, antigenicity, and pathogenicity is essential to thoroughly evaluate the damage caused by the viruses currently circulating in poultry. Therefore, we performed several experiments, including isolating viruses to determine their prevalence, selecting the most predominant virus strains to describe the genetic and antigenic characteristics, and mimicking the complex infection in the field to examine the pathogenicity of LPAIVs in coinfection. Our results facilitate a better understanding of LPAIVs’ evolution, which might be easily neglected owing to their unclear impact on poultry.

Among the AIV subtypes, the H9 and H6 LPAIVs were dominant and detected widely in poultry and wild birds in Asia. Of these, the H9N2 LPAIVs were the most prevalent subtype in China [7,8] and Vietnam [16,17]. All H9 and H6 viruses isolated in this study were phylogenetically close to those previously isolated from poultry in China in 2008 and Vietnam [16,17,18]. This indicates that Vietnamese viruses and those in neighboring countries might share a common ancestor with wild birds. Moreover, most H9 LPAIVs were isolated from chickens, while H6 LPAIVs were mainly isolated from ducks, implying that Vietnamese H9 and H6 LPAIVs have adapted to chickens and ducks, respectively. This phenomenon might be partially explained by the difference in susceptibility to LPAIVs between chickens and ducks due to the difference in receptor distributions in these birds. However, further studies are required to demonstrate the molecular mechanism of this phenomenon. During circulation, the reassortment of H9 and H6 LPAIVs seems to be highly frequent in Vietnam and its neighboring countries [42]. Additionally, as free-grazing ducks are common farming models in Vietnam, repetitive AIV infection in the duck population might occur continuously due to the lack of biosecurity during poultry farming. Conversely, as chickens are kept in a relatively more biosecure environment, this might reduce the risk of multi-infection by AIVs. This specific situation in Vietnam might explain the development of 19 genotypes for representative H6 LPAIVs, while only 6 genotypes are found for representative H9 LPAIVs.

Interestingly, the antigenicity of Vietnamese H9 viruses was similar, forming a single antigenic group. The conserved antigenicity of H9 isolates suggested that these viruses are maintained in the immunologically naïve poultry population in Vietnam despite the high detection of H9 viruses. In contrast, the antigenicity of representative H6 viruses was divided into two different groups, indicating that their antigenicity might have undergone diversification. This phenomenon was also reported in Chinese H6 LPAIVs isolated from 2014 to 2018 [43]. Although the main reason for the antigenic drift in AIV was reported to be selection under pressure from antibodies induced by vaccination, there are no vaccines against H6 LPAIVs in Vietnam. Therefore, the antigenic drift of Vietnamese H6 LPAIVs might be related to natural infection in the domestic duck population. However, further studies are necessary to prove the antigenic drift of H6 viruses in ducks in Vietnam.

Considering the replication of LPAIVs in chickens, previous studies proved that this replication was mainly limited to tissues expressing trypsin, such as epithelial cells [44] in the respiratory and gastrointestinal tracts [45]. Notably, the independent emergence of extrapulmonary replication of H9 LPAIVs with trypsin expression was discovered previously [46,47]. Recently, this phenomenon has been reported in China [48,49], Korea [50], and Egypt [51]. Consistent with these reports, we showed that the representative H9 LPAIV isolated in Vietnam could replicate in extrapulmonary organs, especially in the brain. Interestingly, an insignificant difference in clinical signs was observed regardless of extrapulmonary tissue replication. The H9N2 LPAIV selected for the animal experiment in this study possessed the motif “R-S-S-R” in the HA cleavage site which meets the minimum requirement for the cleavage by furin. However, the furin pathway might not be a major contributor owing to its low efficiency in the cleavability of HA [52]. Therefore, another contributor might exist to support the extrapulmonary replication of H9N2 LPAIVs and further study is necessary to reveal the actual mechanism.

*E. coli* is widely accepted as the most common infectious bacterial pathogen that causes colibacillosis in poultry of all ages [38]. Therefore, a coinfection between this bacteria and the most predominant subtype of AIVs in poultry is highly possible. *E. coli* can obtain virulence factors, such as hydrogen peroxide, nitrous oxide, and various proteases, which might increase its pathogenicity in poultry, and it can thus be considered an APEC [22]. APEC strains lead to systemic or localized damage by causing host cells necrosis, superfluous proinflammatory cytokine release, decreased phagocytosis, and B and T lymphocyte dysfunction, which might promote the replication of AIVs [53]. Coinfections in poultry generate bias and confusion for clinical-based disease identification owing to their complicated clinical picture [54]. Despite the small sample size of bacterial isolates in this study, the number of *E. coli* isolates and their virulence factors might cause severe damage to the poultry industry, especially during coinfection with AIVs. Reports have shown significantly enhanced replication of H9 LPAIV in extrapulmonary tissues in the presence of *E. coli* [53,55,56]. Although the increase in virus titer was observed in many extrapulmonary tissues, significant differences were not observed in the brain, even during coinfection. Moreover, the earlier and higher virus shedding in both the trachea and cloaca of chickens in the coinfection group compared with the single-virus infection indicates a higher impact of the coinfection by a possible synergism between *E. coli* O2 and H9 LPAIV. Conclusively, the coinfection of *E. coli* O2 and H9 LPAIV enhanced the replication and shedding of H9 LPAIV but retained its asymptomatic nature in experimental chickens. This might promote silent spread on a larger scale leading to losses of meat and egg production in the long term, which is the most common impact of LPAIVs. Although the economic losses caused by H9 LPAIVs were prominent, it was difficult to evaluate these viruses precisely and the underlying mechanisms remain unclear. Therefore, long-term studies with appropriate measurements are essential to reveal the mechanisms, mainly focusing on the interaction among pathogens in single infection and coinfection.

Our results suggest that the consecutive evolution of LPAIVs is a major but silent problem. LPAIVs are mainly detected and characterized during active surveillance, which limits the information available on LPAIVs. To overcome this challenge, LPAIVs should be detected during diagnosis. Effective identification of an LPAI outbreak will enable further elucidation of the characteristics of LPAIVs and support better control of AIVs.

## 5. Conclusions

The associations among the genomic data, antigenicity, and pathogenicity of H9 and H6 LPAIVs isolated in Vietnam were thoroughly evaluated. The Vietnamese viruses were found to be genetically related to the previous isolates and those in neighboring countries. The antigenicity of these subtypes was different from that of viruses isolated from wild birds. Antigenicity was more conserved in the H9 viruses than in the H6 viruses. A representative H9 LPAIV exhibited systemic replication in chicken, which was enhanced by coinfection with APEC.

## Figures and Tables

**Figure 1 microorganisms-11-00244-f001:**
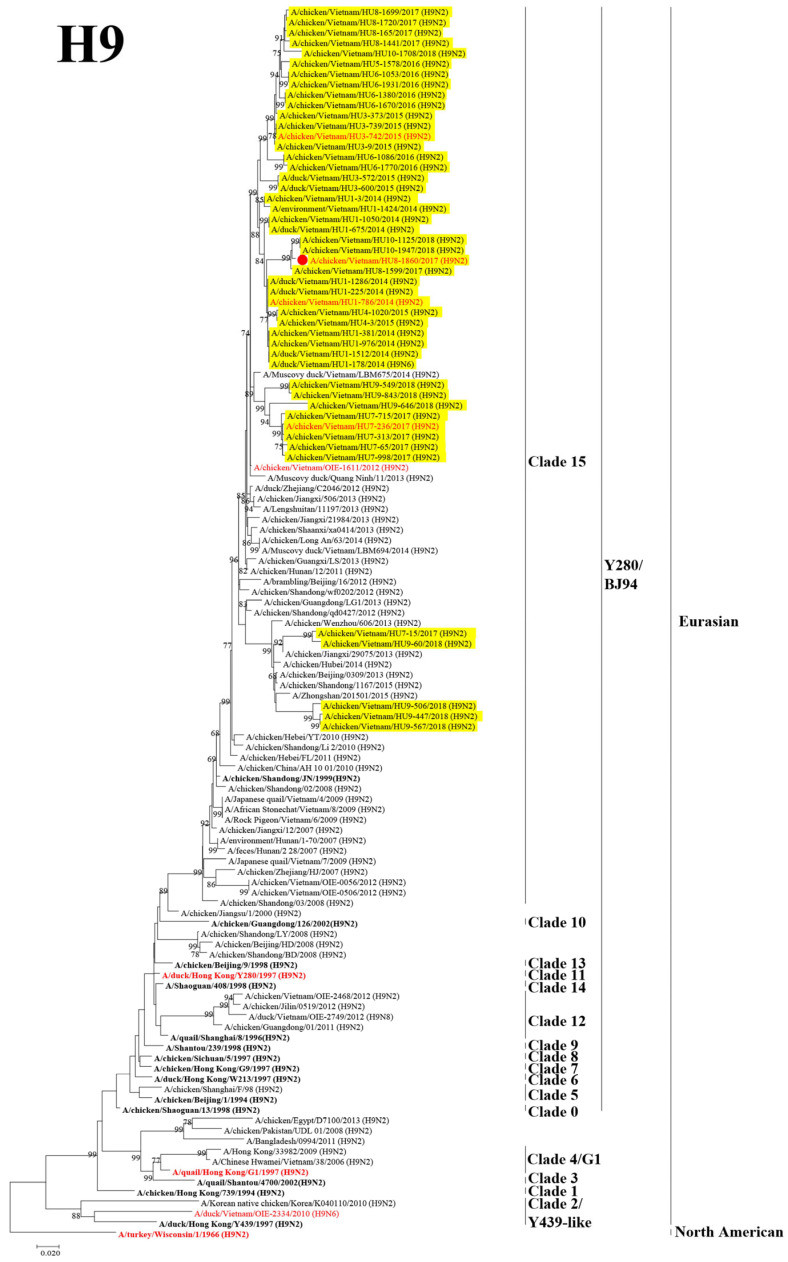
Phylogenetic tree of the HA gene segment of H9 avian influenza viruses. The HA gene segments of the H9 subtype viruses and of the reference strains was analyzed using maximum likelihood (ML) method using MEGA 7.0 software. The numbers at the nodes indicate the probability of the confidence levels from 1000 bootstrap replicates. The H9 viruses in this study are highlighted in yellow, the representative virus of each sublineage/clade is indicated in bold, the viruses selected for antigenic analyses are red highlight, the virus selected for pathogenicity assessment is indicated in the red circle.

**Figure 2 microorganisms-11-00244-f002:**
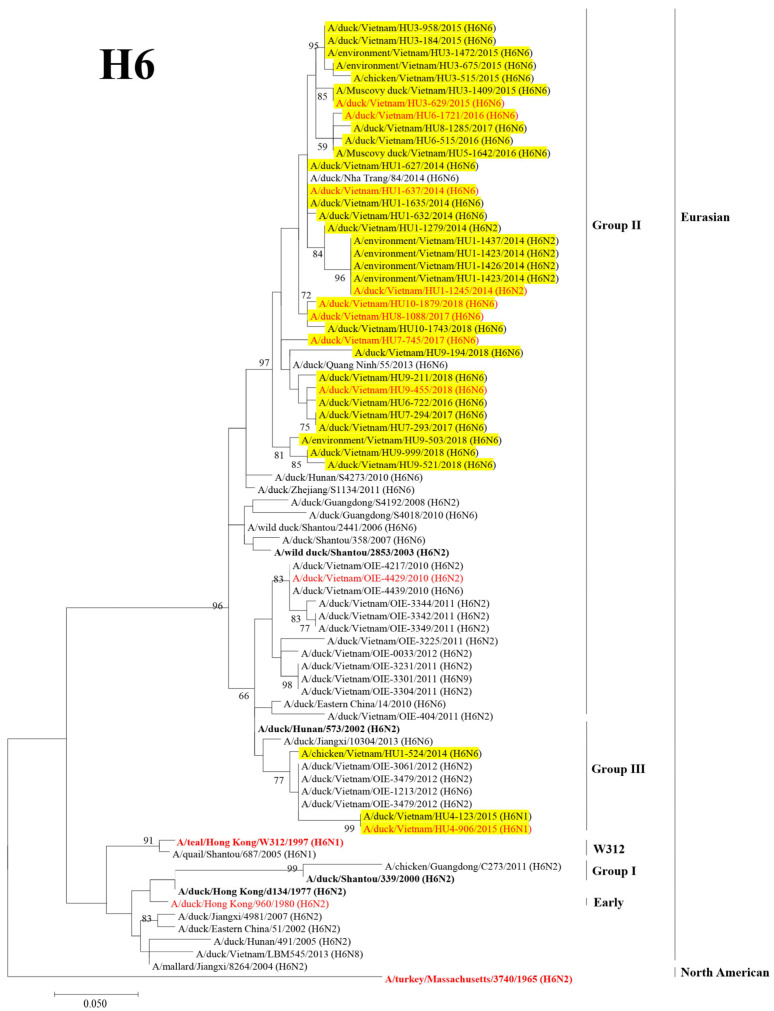
Phylogenetic tree of the HA gene segment of H6 avian influenza viruses. The HA gene segment of the H6 subtype viruses and of the reference strains was analyzed using the ML method using MEGA 7.0 software. The numbers at the nodes indicate the probability of the confidence levels from 1000 bootstrap replicates. The H6 viruses in this study are highlighted in yellow, the representative virus of each sublineage/clade is indicated in bold, and the viruses selected for antigenic analysis are red highlight.

**Figure 3 microorganisms-11-00244-f003:**
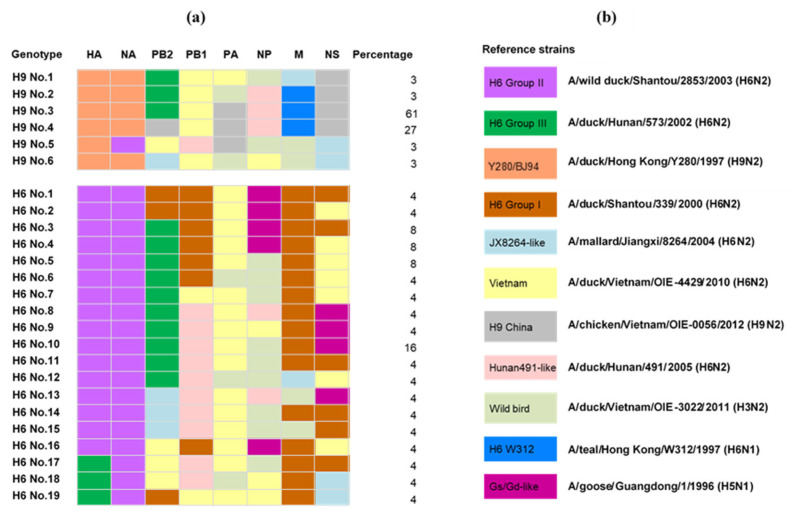
The gene constellations of H6 and H9 avian influenza viruses isolated from poultry in Vietnam. (**a**) Different colors indicate segments whose sequences fall into different major clades clustered. (**b**) The representative strains of phylogenetic analysis.

**Figure 4 microorganisms-11-00244-f004:**
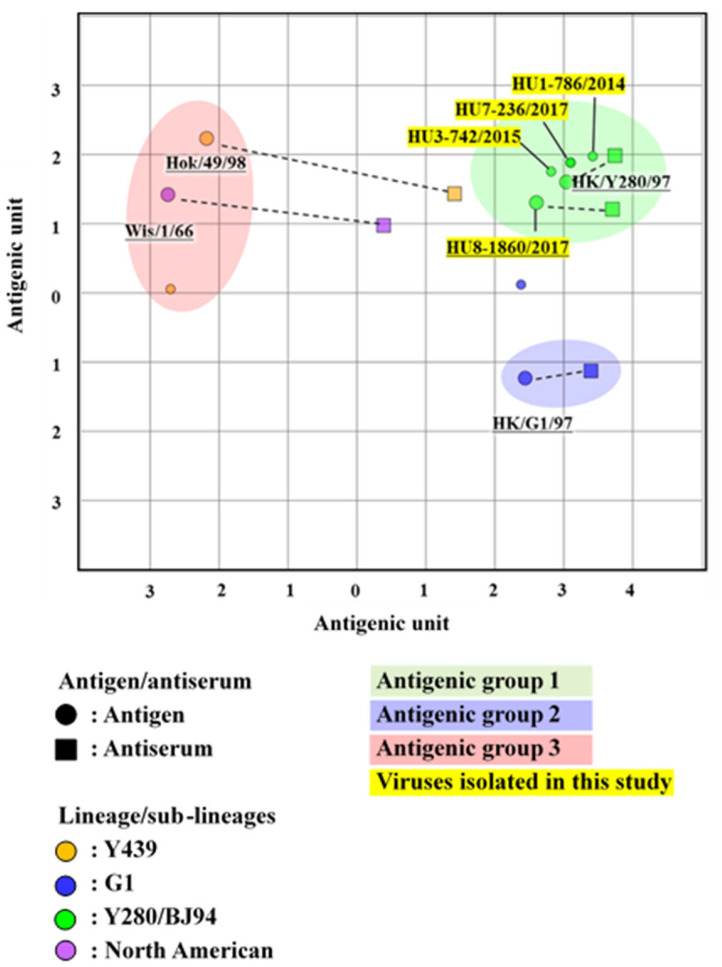
An antigenic map of H9 viruses based on the cross-HI tests on viruses and sera of different lineages. In the antigenic map, both the vertical and horizontal axes represent the antigenic distance. The spacing between the grid lines represents a 1 antigenic-unit distance, corresponding to a 2-fold dilution in the HI assay (2 units correspond to a 4-fold dilution, 3 units correspond to an 8-fold dilution, etc.). Different antigenic groups are indicated by different colors (green, blue, and red). Dotted lines indicate a combination of the homologous viruses and antisera.

**Figure 5 microorganisms-11-00244-f005:**
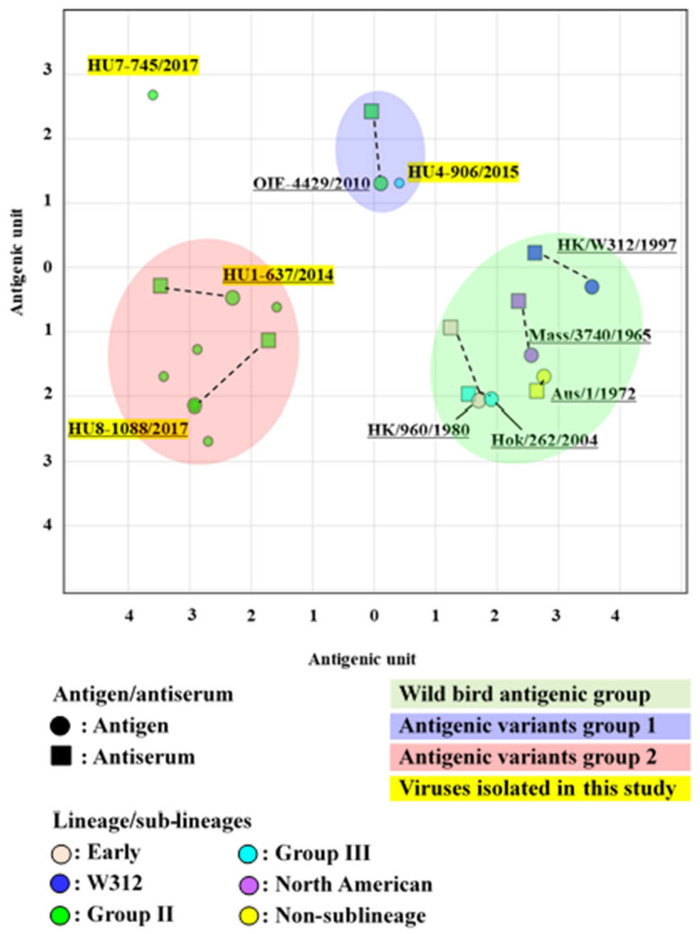
An antigenic map of H6 viruses based on the cross-HI tests on viruses and sera of different lineages. In the antigenic map, both the vertical and horizontal axes represent the antigenic distance. The spacing between the grid lines represents a 1 antigenic-unit distance, corresponding to a 2-fold dilution in the HI assay (2 units correspond to a 4-fold dilution, 3 units correspond to an 8-fold dilution, etc.). Different antigenic groups are indicated by different colors (green, blue, and red). Dotted lines indicate a combination of the homologous viruses and antisera.

**Figure 6 microorganisms-11-00244-f006:**
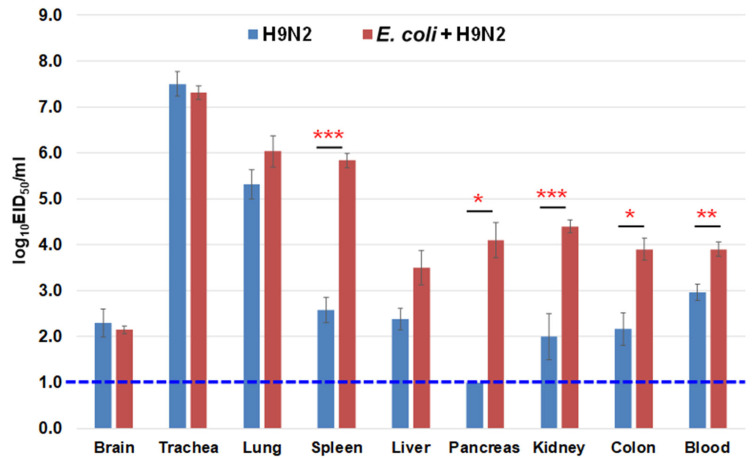
Virus recovery from the internal organs of chickens inoculated with *E. coli* O2 and H9 LPAIV. The chickens were inoculated intranasally with 10^9.0^ CFU of *E. coli* O2 two days prior to challenge with 10^6.0^ EID_50_ of a representative H9 LPAIV (A/chicken/Vietnam/HU8-1860/2017 (H9N2) in the red circle of Figure 1). Virus titers in the organs of chickens collected at 3 dpi. Bars indicate the standard error of mean. Statistical significance was calculated using Tukey’s test. Significant differences in virus titers between groups are indicated by * (*p* < 0.05), ** (*p* < 0.01), and *** (*p* < 0.001). The dotted line indicates the detection limit of the test.

**Figure 7 microorganisms-11-00244-f007:**
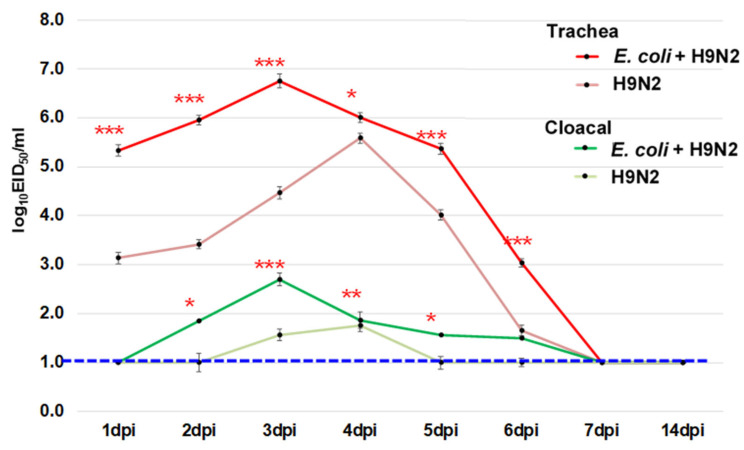
The virus recovery from the trachea and cloacal swabs in chickens. Chickens were inoculated intranasally with 10^9.0^ CFU of *E. coli* O2 two days prior to challenge of 10 ^6.0^ EID_50_ of a representative H9 LPAIV (A/chicken/Vietnam/HU8-1860/2017 (H9N2) in the red circle in Figure 1). The swabs were taken from 1 to 14 dpi. Bars indicate the standard error of mean. Statistical significance was calculated using Tukey’s test. Significant differences in viral titers between groups are indicated by * (*p* < 0.05), ** (*p* < 0.01), and *** (*p* < 0.001). The dotted line indicates the detection limit of the test.

## Data Availability

Sequencing data for viruses in this study were stored in the NCBI database and can be accessed using strain names.

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
