# Peer review of "Genetic, Antigenic, and Pathobiological Characterization of H9 and H6 Low Pathogenicity Avian Influenza Viruses Isolated in Vietnam from 2014 to 2018"

_microorganisms, 2023, doi:10.3390/microorganisms11020244_

Round 1

Reviewer 1 Report

microorganisms-2133150

Le KT, et al.

“Genetic, antigenic, and pathobiological characterization of H9 and H6 low pathogenic avian influenza viruses isolated in Vietnam from 2014 to 2018”

In this manuscript, the authors genetically and antigenically characterized H9 and H6 low pathogenic avian influenza viruses isolated from 2014 and 2018 through active surveillance in Vietnam. The viruses they analyzed were genetically close to the previous Vietnamese isolates and those in neighboring countries. Antigenic analysis indicated that H9 viruses examined were closely related to each other and that H6 viruses examined were largely divided into two groups. In addition, they analyzed the effect of coinfection of H9 virus and avian pathogenic E. coli (APEC) with H9 on the pathogenicity in chickens compared with that of a single infection of H9 virus. The virus titers were higher in the internal organs and blood of chickens in the coinfection group than those in the virus infection group although similar virus titers were seen in the brain and the respiratory organs in chickens in both groups. Also, the virus titers in both trachea and cloacal swabs of chickens in the coinfection group were higher than those in the virus infection group. These results in the coinfection study suggest that the replication of H9 viruses is enhanced when coinfection of H9 virus and APEC occurs. The results presented here are clear and the manuscript is well-written.

Specific Comments

1.           Discussion. p.11 line 440-p.12 line 444. “Conclusively, the coinfection of ….. caused by H9 LPAIVs”. If the infection is asymptomatic, the economic losses might not be increased even if the infection spread silently. Thus, if the authors would like to discuss the economic losses caused by H9 viruses, this sentence should be rephrased.

2.           Results. “3.1. Identification of the AIVs circulating in poultry”. Please describe NA subtypes as well.

3.           Supplementary Figure 1B. There is one isolate (H6N6) in Group III. Thus, the authors should include this isolate in the tree.

4.           Results. “3.5. Antigenic analysis of the H6 LPAIV isolates”. Lines 294-296. “These results implied that ….. from viruses isolated from wild birds”. This might be true. But because the viruses they used are old isolates and/or different sub-lineage, these might affect the results. Thus, the authors should consider these points.

5.           Materials and Methods. “2.6. Animal experiments”. Please describe the reason(s) why chickens were challenged with APEC two days prior to, not on the same day as the virus infection.

Minor points

1.           Lines 72-74. Please cite the reference(s).

2.           Materials and Methods. “2.6. Animal experiments”. Please describe the infection route of viruses and APEC.

3.           Lines 232 and 240. Ref. 37 is not suitable. This might be ref. 17

4.           Line 251. “A total of seven representative H9 viruses”. There are more than seven viruses in Supplementary Table 3. If the representative viruses mean those analyzed in this study, only four viruses are listed in the table. Please clarify and correct this. And “A/chicken/Vietnam/HU1-786/2014” should be bold.

5.           Line 255. “a moderate reaction”. Reactivity is strong compared with homologous virus titer. Thus, this should be modified.

6.           Line 276. “We selected ten representative H6 strains”. This is similar to that described above (no. 4). Only nine viruses are listed in Supplementary Table 4 if the representative viruses mean those analyzed in this study. And “A/duck/Vietnam/HU1-1245/2014” and “A/duck/Vietnam/HU1-637/2014” should be bold.

7.           Figure 4. Items underneath “Antiserum” do not explain each serum. Rather, the same explanation of "Antigen" should be used for "Antiserum" with a different symbol (square). Thus, this figure should be modified properly. And “HU4-786/2015” should be “HU1-786/2014”.

Author Response

Revision Note

 Dear Reviewers,

Thank you very much for your constructive comments and valuable suggestions for improving our manuscript for publication in the Microorganisms. The manuscript, microorganisms-2133150, titled “Genetic, antigenic, and pathobiological characterization of H9 and H6 low pathogenicity avian influenza viruses isolated in Vietnam from 2014 to 2018” was revised according to your suggestions as in follows.

General modifications

  1. Supplementary Table 1 was revised to include the information of the previous studies to provide more details related to the virus isolation results.
  2. We checked grammatical errors throughout the manuscript to select more appropriate words, remove the nonessential, and added explanations in the main text based on reviewers’ suggestions.
  3. The modifications based on comments of reviewers No.1 are highlighted in yellow, the modifications based on comments of reviewers No.2 are highlighted in Cyan, and the modifications in our side are highlighted in green in the revised manuscript.
  4. According to the comments from Editorial office, we ordered grammar check to the company, Enago again for revised manuscript. Since the editing points from the grammar service were so much, we did not use any highlight for these points. However, we will indicate the evidence of re-editing by the company as attached files.

Responses to reviewer No. 1

  1. Major (specific) comments
  2. Discussion. p.11 line 440-p.12 line 444. “Conclusively, the coinfection of ….. caused by H9 LPAIVs”. If the infection is asymptomatic, the economic losses might not be increased even if the infection spread silently. Thus, if the authors would like to discuss the economic losses caused by H9 viruses, this sentence should be rephrased.

Answer:

Thank you very much for your suggestion. We agreed with your suggestion.

Modification:

Page 13, lines 454 – 457: “…leading to losses of meat and egg production in the long term, which is the most common impact of LPAIVs. Although the economic losses caused by H9 LPAIVs were prominent, it was difficult to evaluate these viruses precisely and the underlying mechanisms remain unclear. Therefore long-term studies with appropriate measurements…”

  1. Results. “3.1. Identification of the AIVs circulating in poultry”. Please describe NA subtypes as well.

Answer:

Thank you very much for your suggestion. We agreed with your suggestion.

Modification:

Page 5, lines 205 – 208: “Based on HA classification, 69 of H3, 8 of H4, 213 of H5, 344 of H6, 3 of H7, 698 of H9, 9 of H10, 15 of H11, 1 of H12, and 1 of H13 AIVs were identified, while based on NA classification, 217 of N1, 771 of N2, 2 of N3, 1 of N5, 348 of N6, 7 of N7, 5 of N8, and 10 of N9 AIVs were identified.”

  1. Supplementary Figure 1B. There is one isolate (H6N6) in Group III. Thus, the authors should include this isolate in the tree.

Answer:

Thank you very much for your suggestion. We agreed with your suggestion.

Modification:

Isolate A/chicken/Vietnam/HU1-524/2014 (H6N6) was add in the Supplementary Figure 1B.

  1. Results. “3.5. Antigenic analysis of the H6 LPAIV isolates”. Lines 294-296. “These results implied that ….. from viruses isolated from wild birds”. This might be true. But because the viruses they used are old isolates and/or different sub-lineage, these might affect the results. Thus, the authors should consider these points.

Answer:

Thank you very much for your suggestion. We agreed with your suggestion.

Modification:

Page 9, lines 309 – 311: “These results implied that the H6 viruses isolated in this study have undergone antigenic diversification in the poultry population.”

  1. Materials and Methods. “2.6. Animal experiments”. Please describe the reason(s) why chickens were challenged with APEC two days prior to, not on the same day as the virus infection.

Answer:

Thank you very much for your suggestion. We agreed with your suggestion and the explanation was added in the revised manuscript.

Modification:

Page 4, lines 154 – 158: “E. coli is persistent in the environment and gastrointestinal tract of poultry. This evidence was widely agreed upon in previous studies [36-38]. Thus, the most plausible scenario for E. coli and AIV coinfection in the field is that the poultry were first infected by E. coli and then by AIVs. In this regard, the animal experiment was designed as follows:…”

  1. Minor comments
  2. Lines 72-74. Please cite the reference(s).

Answer:

Actually, this was the reason we conducted this study. The bacteria were isolated in our surveillance and used for this study. And this is the first report that could show strong evidence (we isolated the infectious pathogens for both E. coli and AIVs in the same poultry) about the coinfection of AIVs and E. coli in the fields.

Modification:

No modification.

  1. Materials and Methods. “2.6. Animal experiments”. Please describe the infection route of viruses and APEC.

Answer:

Thank you very much for your clarification. The E. coli and H9 LPAIVs were inoculated into experimental chickens by the intranasal route and this explanation was added in the revised manuscript.

Modification:

Page 4, lines 160 – 161: “E. coli and H9 LPAIVs were inoculated into experimental chickens intranasally.”

  1. Lines 232 and 240. Ref. 37 is not suitable. This might be ref. 17.

Answer:

Thank you very much for your careful review. We agreed and revised the reference following your suggestion.

Modification:

Page 7, lines 239: “…as previously described [16,17].”

Page 8, lines 242: “…were classified based on previous studies [16,17].”

  1. Line 251. “A total of seven representative H9 viruses”. There are more than seven viruses in Supplementary Table 3. If the representative viruses mean those analyzed in this study, only four viruses are listed in the table. Please clarify and correct this. And “A/chicken/Vietnam/HU1-786/2014” should be bold.

Answer:

Thank you very much for your suggestion. Herein we would like to mention the Vietnamese isolates therefore we count a total of seven representative H9 viruses. To avoid confusion, we added the word “Vietnamese” was added in the main text as well as the format in Supplementary Table 3 following your suggestion.

Modification:

Page 8, lines 261 – 263: “To examine changes in the antigenicity of H9 viruses, isolates from 2009 to 2012 were antigenically analyzed using the cross-HI test. Seven Vietnamese representative H9 viruses were analyzed (Table S3)”

Supplementary Table 3: A/chicken/Vietnam/HU1-786/2014 was bolded.

  1. Line 255. “a moderate reaction”. Reactivity is strong compared with homologous virus titer. Thus, this should be modified.

Answer:

Thank you very much for your intensive inquiry. We agreed with your suggestion and the explanation was revised in the main text following your suggestion.

Modification:

Page 8, lines 265 – 269: “Similarity, all Vietnamese isolates showed a strong reaction with the antisera of Hok/49/98 (H9N2) (Y439 sublineage). However, these viruses showed a moderate re-action with the antisera of HK/G1/97 (H9N2) (G1 sublineage) and weak reaction with the antisera of Wis/1/56 (H9N2) (North American lineages)”

  1. Line 276. “We selected ten representative H6 strains”. This is similar to that described above (no. 4). Only nine viruses are listed in Supplementary Table 4 if the representative viruses mean those analyzed in this study. And “A/duck/Vietnam/HU1-1245/2014” and “A/duck/Vietnam/HU1-637/2014” should be bold.

Answer:

Thank you very much for your suggestion. Same as comment No.4, we would like to mention the Vietnamese isolates therefore ten Vietnamese representative H6 viruses were selected. To avoid confusion, we added the word “Vietnamese” was added in the main text as well as the format in Supplementary Table 4 following your suggestion.

Modification:

Page 9, lines 288 – 290: “Isolates obtained in 2010 were used for the cross-HI test to evaluate the antigenic drift. Ten Vietnamese representative H6 strains were selected for antigenic analysis …”

  1. Figure 4. Items underneath “Antiserum” do not explain each serum. Rather, the same explanation of "Antigen" should be used for "Antiserum" with a different symbol (square). Thus, this figure should be modified properly. And “HU4-786/2015” should be “HU1-786/2014”.

Answer:

Thank you very much for your careful review. Your suggestion is very valuable to improve the explanation of the figure. We applied for both Figures 4 and 5 in the revised manuscript.

Modification:

The content of Figure 4 as well as the legend of Figures 4 and 5 were revised.

Reviewer 2 Report

In this study, the authors evaluated the genomic data, antigenicity, and pathogenicity of H9 and H6 low pathogenic avian influenza viruses isolated from poultry in Vietnam in 2014-2018. A thorough genetic analyses of a large number of isolates was performed, thus providing additional data which is important for routing surveillance of avian influenza viruses in this region. The antigenic analyses of H6 and H9 isolates, as well as pathogenicity studies of H9 virus alone or in co-infection with E.coli strain in chickens also have an added value. In general, manuscript is well-written and contains the data that support the authors’ conclusions. However, some points need clarificationÑŽ

Major

1.       Paragraph 3.1 refers to the Table S1 where the isolates were already reported in other papers. The authors should clearly describe what isolates were new to these studies or properly refer to the other previously published research. In addition, not all the references from Table S1 are cited in the References list of the main manuscript.

2.       The choice for the schedule of viral-bacterial co-infection should be clearly explained.

Minor

1.       Lane 121. “to construct the phylogenetic tree”?

2.       Figures 1 and 2 may be drawn in color for better presentation

3.       It would be beneficial if the authors added an Illustrative material for paragraph 3.6.

The E.coli and H9N2 strains used for chicken studies are provided only in Figures 6-7 legends. This information should be given in the main text as well.

Author Response

Revision Note

Dear Reviewers,

Thank you very much for your constructive comments and valuable suggestions for improving our manuscript for publication in the Microorganisms. The manuscript, microorganisms-2133150, titled “Genetic, antigenic, and pathobiological characterization of H9 and H6 low pathogenicity avian influenza viruses isolated in Vietnam from 2014 to 2018” was revised according to your suggestions as in follows.

General modifications

  1. Supplementary Table 1 was revised to include the information of the previous studies to provide more details related to the virus isolation results.
  1. We checked grammatical errors throughout the manuscript to select more appropriate words, remove the nonessential, and added explanations in the main text based on reviewers’ suggestions.
  2. The modifications based on comments of reviewers No.1 are highlighted in yellow, the modifications based on comments of reviewers No.2 are highlighted in Cyan, and the modifications in our side are highlighted in green in the revised manuscript.
  3. According to the comments from Editorial office, we ordered grammar check to the company, Enago again for revised manuscript. Since the editing points from the grammar service were so much, we did not use any highlight for these points. However, we will indicate the evidence of re-editing by the company as attached files.

Responses to reviewer No. 2

  1. Major comments
  2. Paragraph 3.1 refers to the Table S1 where the isolates were already reported in other papers. The authors should clearly describe what isolates were new to these studies or properly refer to the other previously published research. In addition, not all the references from Table S1 are cited in the References list of the main manuscript.

Answer:

Thank you very much for your kind suggestion. We agreed with your suggestion and added information about the previous studies related to this surveillance. In addition, we clarified which part of the surveillance was mentioned in the previous studies.

Modification:

Page 5, lines 201 – 205: “The virological characterization were conducted for some H6 and H9 viruses isolated in Hue province in 2014, H7 viruses isolated in Vinh Long province in 2018, and H5 HPAIVs isolated from 2014 to 2017 in previous studies [18, 29, 40]. The other isolates were reported only in an epidemiological aspect [19, 41] or newly characterized in the present study.”

  1. The choice for the schedule of viral-bacterial co-infection should be clearly explained.

Answer:

Thank you very much for your suggestion. We understood and agreed with your concern. The explanation was added to clarify the reason for the selection of a viral-bacterial co-infection schedule.

Modification:

Page 4, lines 154 – 158: “E. coli is persistent in the environment and gastrointestinal tract of poultry. This evidence was widely agreed upon in previous studies [36-38]. Thus, the most plausible scenario for E. coli and AIV coinfection in the field is that the poultry were first infected by E. coli and then by AIVs. In this regard, the animal experiment was designed as follows:…”

  1. Minor comments
  2. Lane 121. “to construct the phylogenetic tree”?

Answer:

Thank you very much for your suggestion. We agreed with your suggestion and revised in the main text.

Modification:

Page 3, lines 123: “…construct the phylogenetic tree using MEGA 7.0 software [27].”

  1. Figures 1 and 2 may be drawn in color for better presentation

Answer:

Thank you very much for your suggestion to improve our manuscript. We agreed with your suggestion and prepare the color version of Figures 1 and 2.

Modification:

Figures 1 and 2 now are colored and the legend has also been modified.

  1. It would be beneficial if the authors added an Illustrative material for paragraph 3.6.

Answer:

Thank you very much for your suggestion. Supplementary table 2 included the material and results of this section. We succeed in reproducing the experiment to figure out the virulence-associated genes of APEC described by Wang et al., 2014; Janßen et al. 2001; Subedi et al. 2018; Prioste et al. 2013; and Horne et al. 2000 without any difficulty. Furthermore, the APEC was a minor component of this surveillance and we planned to expand this part in the future. Therefore, APEC research was a pilot study with limited information. In the future project, we will prepare and describe more about this point.

Modification:

No modification.

  1. The E. coli and H9N2 strains used for chicken studies are provided only in Figures 6-7 legends. This information should be given in the main text as well.

Answer:

Thank you very much for your suggestion. We agreed with your suggestion and revised in the main text.

Modification:

Page 10, lines 341: “…viral infection group (A/chicken/Vietnam/HU8-1860/2017 - H9N2 alone)…”

Page 10, lines 344 – 345: “…coinfection group (E. coli O2 and A/chicken/Vietnam/HU8-1860/2017 - E. coli + H9N2)…”

Page 10, lines 358 – 360: “To determine the difference in virus shedding between the virus infection (A/chicken/Vietnam/HU8-1860/2017 - H9N2 alone) and coinfection groups (E. coli O2 and A/chicken/Vietnam/HU8-1860/2017 - E. coli + H9N2)…”

Round 2

Reviewer 2 Report

The authors revised their manuscript according to the reviewers' comments and the paper now can be accepted for publication